# Topic-Conversation Relevance (TCR) Dataset and Benchmarks

**Yaran Fan**
Microsoft
yaran.fan@microsoft.com

**Jamie Pool**
Microsoft
jamie.pool@microsoft.com

**Senja Filipi**
Microsoft
sefilipi@microsoft.com

**Ross Cutler**
Microsoft
ross.cutler@microsoft.com

## Abstract

Workplace meetings are vital to organizational collaboration, yet a large percentage of meetings are rated as ineffective [1]. To help improve meeting effectiveness by understanding if the conversation is on topic, we create a comprehensive Topic-Conversation Relevance (TCR) dataset that covers a variety of domains and meeting styles. The TCR dataset includes 1,500 unique meetings, 22 million words in transcripts, and over 15,000 meeting topics, sourced from both newly collected Speech Interruption Meeting (SIM) data and existing public datasets. Along with the text data, we also open source scripts to generate synthetic meetings or create augmented meetings from the TCR dataset to enhance data diversity. For each data source, benchmarks are created using GPT-4[1] to evaluate the model accuracy in understanding transcription-topic relevance.

## 1  Introduction

Since the 2020 COVID-19 pandemic, an increasing share of meetings have shifted from in-person to online. The Gartner 2021 Digital Worker Experience Survey reports that the number of in-person meetings dropped from 63% in 2019 to 33% in 2021 [2]. The same survey predicted that in 2024, only 25% of the meetings will happen in person.

Together with the growing number of online meetings, there are ongoing complaints about ineffective meetings due to a lack of focused discussions or focused tasks [3, 4, 5, 6]. Having a meeting facilitator to keep the discussions focused is one of the meeting design characteristics enabling more effective meetings [6, 7].

Measuring how relevant a conversation transcript is to an intended topic is crucial to quantifying how focused the communication is, and to creating tools that behave as a virtual meeting moderator by keeping the discussion on-track. A very low rating on the relevance of the conversation to the topic meant for discussion would be an indicator of a non-focused discussion. In practice this translates to the problem of keeping discussions focused on a predefined meeting agenda.

While there is existing work about the importance of topics serving as input to text summarizing models [8, 9], we could not find references about work studying the relevance of a topic to a particular body of text it didn't originate from. One of our intuitions for why this field has had little exploration is because of technological limitations before the recent Generative AI advancements.

---

[1]The GPT-4 model used in this paper is GPT-4-32k.

38th Conference on Neural Information Processing Systems (NeurIPS 2024) Track on Datasets and Benchmarks.

With the current advancement in the field of Generative AI, deep understanding of language and relationships between bodies of text has reached new levels of accuracy, and has gotten very close to human performance [10, 11, 12, 13, 14].

To begin investing in measuring the relevance of a conversation to a predefined agenda topic, a comprehensive dataset of conversations associated with the topic of each conversation section is needed. Ideally, the topics should be defined before the conversation starts in a pre-meeting agenda style. There are several public datasets built from real human conversations that serve as the base for our Topic-Conversation Relevance (TCR) dataset; however, most of the topics from these datasets are post-meeting minutes that summarize what happens instead of what is planned.

The contributions of the TCR dataset are (1) We create a large topic-conversation dataset covering multiple domains of meetings. This dataset consists of the newly collected meetings and aggregated public data sources. (2) We use GPT-4 to rewrite long and detailed meeting minutes into a pre-meeting agenda topic style. (3) We provide a design of an extensible schema that allows users to create variations of meetings where topics can be flexibly added and removed. (4) We open source scripts for data augmentation and synthetic meeting creation on top of the TCR dataset.

We review the related works in Section 2. We present the datasets and the schema in Section 3, and elaborate to discuss the new SIM data collection and public data sources. In Section 4 we go over the benchmark results generated by GPT-4 on the Topic-Conversation Relevance task, and share insights from running such prompts across datasets. In Section 5, we describe limits and future work in this direction.

## 2   Related Work

In this paper, we refer to "topics" as the key points to be discussed during a meeting and such topics would have been put in the meeting agenda by the organizer before the meeting starts. To the best of the authors knowledge there is no research related to the task of measuring conversation relevance to pre-meeting agenda topics. However, the related topic of meeting summarization, or minuting has been well studied.

Two challenges (AutoMin) in the field of meeting summarization have been held where teams participated in order to progress the field [15, 16]. The first challenge had teams using BART-based models achieving the best performance [17, 18]. With the improvements in generative AI and the growing adoption of Large Language Models (LLMs), a second challenge was done recently. In this challenge, the participants [19, 20, 21] achieved good results with different large models, such as, Llama-based Vicuna [22], Dolly [23], and GPT-3's text-davinci-003 [24]. The challenge organizers used GPT-4 for benchmarking as well and it demonstrated good performance for the meeting summarization task. The organizers also used GPT-4 to evaluate submissions along with human evaluation results, but found that it was unreliable for this task. The challenge organizers also called out the need to answer research questions related to transcript summary relevance, to better understand content and coverage from different annotators.

There are multiple datasets for the task of benchmarking the meeting summarization task. Most of such individual datasets often contains only one type of meeting. The AMI dataset [25] is a collection of meetings transcripts and summaries that cover the topic of product design in an scenario setting and a small amount of non-scenario meetings. The topic annotation is very brief and limited. The ICSI [26] Corpus contains 75 project meetings and discussions in an academic environment. It has high-level human-annotated topics that are very brief. MeetingBank [27] is a dataset of 1,250 city council meetings from multiple US counties. Detailed meeting minutes for each meeting subsection are documented in this dataset. The QMSum [28] dataset aggregates three public data sources (ICSI, AMI, and Parliament meetings from Welsh and Canada) and generates minutes for the text summarization tasks. The paper further shows that for a BART model that training a model on data from one of the datasets and testing it on the other one leads to poor performance. By training on all datasets they were able to build a more robust model. To further expand on data for the automatic minute task Nedoluzhko [29] put together the ELITR data corpus. This data consists of meetings in both Czech and English, with transcripts and meeting minuting being taken by different annotators. In order to align the transcripts with the minutes the tool ALIGNMEET [30] was used.

Table 1: Topic-Conversation Relevance (TCR) Dataset Statistics

| Category | Data Name | Number of Meetings | Number of Topics | Words | Duration (Hours) |
|---|---|---|---|---|---|
| **Unique Meetings** | SIM | 84 | 84 | 529,012 | 48.6 |
| | SIM_syn100 | 100 | 348 | 500,825 | 45.7 |
| | ICSI | 75 | 489 | 767,437 | 70.4 |
| | MeetingBank | 1,100 | 6,595 | 19,626,469 | 2,493.8 |
| | NCPC | 20 | 160 | 423,305 | 47.2 |
| | QMSum_AMI | 96 | 510 | 489,961 | 54.4 |
| | QMSum_Parliament | 20 | 158 | 276,620 | 30.7 |
| | ELITR | 11 | 94 | 56,521 | 6.3 |
| | **Sub Total** | **1,506** | **8,438** | **22,670,150** | **2,797** |
| **Different Annotations** | QMSUm_ICSI | 52 | 288 | 527,206 | 48.8 |
| | MeetingBank ReAnnotated | 1,100 | 6,585 | 19,626,469 | 2,493.8 |

Table 2: Balanced Topic-Conversation Relevance (TCR) Dataset Statistics

| Data Name | Number of Meetings | Number of Topics | Words | Duration (Hours) |
|---|---|---|---|---|
| SIM | 84 | 84 | 529,012 | 48.6 |
| SIM_syn100 | 100 | 348 | 500,825 | 45.7 |
| ICSI | 75 | 489 | 767,437 | 70.4 |
| QMSUm_ICSI | - | 288 | - | - |
| QMSum_AMI | 96 | 510 | 489,961 | 54.4 |
| QMSum_Parliament | 20 | 158 | 276,620 | 30.7 |
| MeetingBank_rnd30 | 30 | 189 | 461,155 | 58.0 |
| MeetingBank_ReAnnotated_rnd30 | - | 189 | - | - |
| NCPC | 20 | 160 | 423,305 | 47.2 |
| ELITR | 11 | 94 | 56,521 | 6.3 |
| **Total** | **436** | **2,509** | **3,504,836** | **361** |

## 3 Topic-Conversation Relevance (TCR) Dataset

We create the TCR Dataset that covers a variety of meeting topics and styles. The dataset consists of both meeting data collected by the author team and existing publicly available datasets.

Overall, the TCR dataset contains 1,506 unique meetings, 22 million words in transcripts and more than 15,000 meeting topics. Table 1 provides high-level statistics of the dataset.

The pre-selected MeetingBank data is large comparing with other data sources. To balance the meeting styles and create representative benchmark results, we also create a subset of 30 randomly selected meetings denoted as `MeetingBank_rnd30`. The subset is available separately from the complete MeetingBank data in the TCR dataset. A summary of the balanced subsets is presented in Table 2.

We also provide exploratory analysis of per meeting metrics in Table 3. The full exploratory analysis with standard deviations is provided in the Appendix 5 . The dataset and related scripts are available in the topic_conversation GitHub repository[2].

### 3.1 Data Schema

All data files in the TCR dataset are in JSON format. An example schema is presented in Figure 1.

Data from different sources are split into separate files. In each file, data is grouped by meeting. For each meeting, we provide meeting level metadata and detailed topic level information. The topics

---

[2]Repository topic_conversation: `https://github.com/microsoft/topic_conversation`

Table 3: Exploratory Analysis of Mean Metrics per Meeting by Data Source

| Data Source | Meeting Duration (minutes) | N Speakers per Meeting | N Topics per Meeting | Topic Duration (minutes) | Topic Text Lenth (words) |
|---|---|---|---|---|---|
| SIM | 34.24 | 4.00 | 1.00 | 34.24 | 11.32 |
| SIM_syn100 | 27.24 | 4.00 | 3.48 | 7.83 | 10.72 |
| ICSI | 45.19 | 6.20 | 6.52 | 6.93 | 2.85 |
| QMSum_ICSI | 44.88 | 6.31 | 4.27 | 10.40 | 4.15 |
| QMSum_AMI | 34.03 | 4.00 | 3.94 | 8.45 | 6.76 |
| QMSum_Parliament | 92.21 | 23.80 | 6.45 | 13.90 | 8.43 |
| MeetingBank | 109.86 | 8.54 | 5.98 | 18.36 | 59.47 |
| MeetingBank_ReAnnotated | 109.86 | 8.54 | 5.97 | 18.39 | 10.03 |
| NCPC | 141.69 | 25.60 | 8.00 | 17.71 | 6.27 |
| ELITR | 34.26 | 5.45 | 7.64 | 4.01 | 6.95 |

are ordered by start time. For each topic, the corresponding transcripts are presented in a list. Each transcript line contains the raw contents in text, speaker ID, time information, line and word counts. If the original data source does not have timestamps, the time information is estimated based on word counts at a fixed 150 words per minute rate for each transcript line. In such cases, the metadata marks "timestamp_source" as "estimated" for the entire meeting.

We provide two scripts (*script_create_synthetic_meetings_SIM.py, script_augment_data.py*) in the project repo to create more synthetics meetings or generate augmented version of the existing datasets. Outputs from those scripts follow the same data schema and can be easily combined into the existing TCR dataset.

### 3.2 New Data Collection: Speech Interruption Meetings (SIM)

In a previous study done by the team regarding speech interruptions and meeting inclusiveness [31], we collect multi-party online meetings in which participants actively interact with each other to debate a topic. This Speech Interruption Meetings (SIM) dataset is released for the first time as part of the TCR dataset. We also create 100 synthetic meetings on top of these raw meetings. Both the original meetings and synthetics meetings are included in the TCR dataset.

#### 3.2.1 Raw Data

In total, we include 84 raw meetings (48.6 hours) in the TCR dataset. The meetings cover 14 different topics and there are about 530,000 words in the transcripts. In total, 149 unique speakers[3] participated in this batch of data collection. Speaker distribution details can be found in Appendix Table 6.

The SIM data captures natural online meeting dynamics. To collect the data, we invite 4 participants to join a remote conference call on Microsoft Teams. Each meeting has a single dedicated topic that can elicit debate. The participants discuss the topic for about 30 minutes. Natural interactions between participants are strongly encouraged. We collect separate audio channels and machine-generated transcripts for each meeting. In the transcripts, the participants are marked as *speaker_1,2,3,4* randomly within each meeting. We only include transcripts data in the TCR dataset at this stage as audio is not directly related to the the benchmark task.

#### 3.2.2 Synthetic Meetings

Given the raw meetings from the SIM dataset has only one dedicated topic per meeting, we also generate 100 synthetic meetings with multiple topics by randomly combining meetings snippets from different topics together.

The workflow to generate such synthetic meetings involves the following steps. First, we remove the first and last 5 minutes of the transcripts, to eliminate potential meeting setup contents, greetings, and icebreaker talk. These trimmed meetings are the candidate meetings. Second, for each new synthetic

---

[3]Participant consent: each participant signed a consent form covering data usage and release.

```
{
    "EXAMPLE_DATA_SOURCE": {   # Data Source Name
        "EXAMPLE_MEETING_NAME": { # Meeting Name
            "metadata": { # Metadata
                "topic_annotation_source": "EXAMPLE_ANNOTATION_SOURCE", # Annotation Source of Topics
                "timestamp_source": "EXAMPLE_TIMESTAMP_SOURCE", # Annotation Source of Timestamps
                "meeting_start_s": 0.0, # Start Time of Meeting in Seconds
                "meeting_end_s": 1800.0, # End Time of Meeting in Seconds
                "meeting_start_line": 0.0, # Start Line of Meeting in Transcripts
                "meeting_end_line": 200.0, # End Line of Meeting in Transcripts
                "meeting_trans_word_count": 3000.0, # Total Word Count of Meeting Transcripts
                "variations": {} # Type of variations included in the meeting.
                               # Refer to 'script_augment_data.py' for creating meeting variations.
            },
            "topics": { # Topics of the Meeting
                "EXAMPLE TOPIC TEXT 1": { # Topic content
                    "topic_start_s": 0.0, # Start Time of Topic in Seconds
                    "topic_end_s": 500.16, # End Time of Topic in Seconds
                    "topic_start_line": 0.0, # Start Line of Topic in Transcripts
                    "topic_end_line": 77.0, # End Line of Topic in Transcripts
                    "topic_trans_word_count": 900.0, # Total Word Count of Topic Transcripts
                    "transcripts": [ # Transcripts of the Topic
                        {
                            "line_id": 0.0, # Line ID/Number
                            "speaker": "speaker_3", # Speaker Name
                            "start_s": 0.0, # Start Time of Transcript in Seconds
                            "end_s": 3.65, # End Time of Transcript in Seconds
                            "contents": "Hello everyone, welcome to today's talk.", # Transcript Content
                            "word_count": 6.0, # Word Count of Transcript
                            "cum_wc": 6.0 # Cumulative Word Count of Transcript
                        },
                        {
                            # Next Line of Transcript and Related Information with the same format
                        },
                        # ...
                    ]
                },
                "EXAMPLE TOPIC TEXT 2": {
                    # Next Topic and Transcripts with the same format
                }
            },
        }
    }
}
```

Figure 1: Topic-Conversation Relevance (TCR) Dataset Schema

meeting, we randomly decide how many unique topics (2 to 5) to include. Then, for each unique topic, we randomly select 5 to 11 minutes consecutive transcripts from candidate meetings with that topic. Based on the setup, the generated 100 synthetic meetings have an average meeting length of 28 minutes. We refer to this set as SIM_syn100.

The scripts that we use to generate the SIM_syn100 data is shared in the project repo. With the configurable parameters, users can create an arbitrary number of synthetic meetings with the desired number of topics and duration splits.

### 3.3 Public Data Sources

To make the TCR dataset cover a wide range of meeting styles and domains, we integrate another 5 publicly available data sources. In this section, we describe the pre-processing procedures we apply to each of them.

#### 3.3.1 ICSI Corpus

We use all 75 meetings from the the ICSI Corpus [26]. Starting from the word-level transcripts from the original corpus, we exclude the tags for non-verbal events and keep only the transcribed words. This is because for real-time machine-generated transcripts, such events are not marked as tags, but either transcribed as part of the contents, or omitted. For long utterances from the same speaker, we assign a line break in the transcript either when an end-of-sentence tag occurs, or there is a gap that is at least 0.5 second long between two words. We assign timestamps for each sentence based on the original word-level timestamps from the data source.

We make minor adjustments to the topic annotations if there is an identify-mismatch problem between the topics and the speaker IDs. The speaker IDs for each meeting are assigned as speaker_A,

speaker_B, etc., however, the topic annotations refer to speakers by either their metadata ID (e.g., me001) or their first names. To align the different identify systems, we refer to the metadata and convert the speaker reference style in the topic annotations to align with the transcripts by using speaker IDs. This guarantees the data consistency within annotations.

### 3.3.2 Selected QMSum Dataset

The QMSum [28] data has 3 different input data sources and we treat them separately. For QMSum_ICSI data, we use the pre-processed transcripts and timestamps from the original ICSI corpus. We use the QMSum annotations as the new topics. Given the topic styles and the section breaks are very different between QMSum_ICSI and the original ICSI Corpus, we decide to keep both sets of meetings and create benchmark results for them separately. For QMSum_AMI and QMSum_Parliament data, we remove non-verbal tags from the transcripts. As timestamps are not available in QMSum, we create estimated timestamps by the fixed 150 words per minute rate for these two data sources.

The annotations are done as meeting minutes in the QMSum dataset. In cases where the transcripts are not included in the minuting, we fill the empty values by the following logic. If the missing topic is at the very beginning of the meeting, we assign a topic of "Beginning_no_topic"; if the missing topic is at the very end of the meeting, we assign a topic of "Ending_no_topic". If the lack of annotation happens between two topics, we assume the previous topic continues and fill the empty value by taking the previous topic. In the QMSum annotation, it is also possible that one line of transcripts belong to multiple topics. We use the first annotation based on the timestamps. In any given meeting, if more than 15% of the transcripts have missing annotations or overlapping annotations issues, we exclude the meeting due to undesirable annotation quality. Overall, we keep 168 out of the original 232 meetings.

### 3.3.3 Selected MeetingBank Dataset

We use the timestamps in the metadata from the original data source to exclude meetings that are shorter than 15 minutes. In total, 1,100 out of the 1,250 MeetingBank [27] meetings are included in our dataset. We remove unicode from both the transcripts and annotations. Though some of the original timestamps do not start from 0, we keep the original timestamps as it is necessary to locate the corresponding audio contents if needed. In the TCR data, it is very easy to align the beginning to 0 by removing the start timestamp documented in the meeting metadata.

In the TCR dataset, we provide two sets of annotations for the MeetingBank data:

**Original Annotations**   We take the "summary" field from the MeetingBank metadata as the topic annotations. These annotations are in meeting minutes styles and often are long and very detailed. If in the original data source one sentence belong to multiple summaries, we keep only the first occurrence.

**Re-annotated Topics**   The original meeting summaries contain not only the topic for a section but often the outcomes. To have pre-meeting style topics, we need to remove outcomes that would not have been known before the meeting happens. Additionally some of the meeting minutes are excerpts from the transcripts, so modifying the annotations would give a more accurate representation of the topic-conversation relevance benchmark. In order to rewrite a summary to a pre-meeting agenda type of topic, a GPT-4 prompt is developed. An example of the input and output is shown below:

- Original summary: *A bill for an ordinance changing the zoning classification for 5611 East Iowa Avenue in Virginia Village. Approves an official map amendment to rezone property located at 5611 East Iowa Avenue from S-SU-D to S-RH-2.5 (suburban, single-unit to suburban, rowhouse) in Council District 6. The Committee approved filing this item at its meeting on 7-10-18.*
- Re-annotated topic: *Zoning Change for 5611 East Iowa Avenue in Virginia Village.*

To guarantee high quality of the re-annotated topics, we randomly selected 100 samples and collected Mean Opinion Score (MOS) scores of 1-5 from 3 project members. A score of 5 means that the re-annotated topic is in a proper pre-meeting agenda style and it captures the key information from

```
{
    "EXAMPLE_AUG_DATA": {  # Data Source Name
        "EXAMPLE_AUG_MEETING": { # Meeting Name
            "metadata": { # Metadata
                # Same metadata structure including data source, time and transcripts information
                # ...
                "variations": {
                    "variation_addToics": [ # Added topic list
                        "ADDED TOPIC A1 TEXT"],
                    "variation_removeTopics": [ # Removed topic list
                        "REMOVED TOPIC R1 TEXT"]
                }
            },
            "topics": { # Topics of the Meeting
                # Topic and Transcripts that are kept
                # ...
                "ADDED TOPIC A1 TEXT": {# Added planned topic
                    "topic_start_s": -1,
                    "topic_end_s": -1,
                    "topic_start_line": -1,
                    "topic_end_line": -1,
                    "topic_trans_word_count": 0,
                    "transcripts": []
                },
                "REMOVED TOPIC R1 TEXT": { # Remove topic and contents
                    # Contents of TOPIC R1
                },
            },
        }
    }
}
```

Figure 2: Metadata Schema for Augmented Meetings

the original annotation. A score of 1 means that neither is true. Across the 300 votes, the average MOS is 4.6 and the median is 5.

The full MeetingBank data with re-annotated topics is referred to as `MeetingBank_ReAnnotated` and the subset with the same 30 meetings is referred to as `MeetingBank_ReAnnotated_rnd30`.

### 3.3.4 Selected NCPC Meetings

The National Capital Planning Commission (NCPC) [32] is a government agency that meets once a month to discuss projects for in the united states capitol region. The meeting agendas and transcripts are publicly available. To the best of our knowledge, the TCR dataset is the first work to add this data source to a structured dataset. We randomly select 20 NCPC meetings where agenda is available. Both meeting transcripts and agenda topics are documented in the same PDF file for each meeting. In order to convert the data to a structured format, the PDFs are converted to text files, and the body of the text is extracted, along with the topic titles. As the PDFs do not share the same structure, additional manual adjustments are applied to guarantee a high conversion accuracy. The original transcripts do not have time information, hence the timestamps are estimated with the fixed rate of 150 words per minute.

### 3.3.5 Selected ELITR Dataset

The ELITR [29] data is a corpus of meetings in Czech and English containing transcripts along with minutes written by multiple annotators. As our work focuses on English only at this stage, we keep just the English meetings. Among the English meetings, 49 have meeting minutes that can be aligned with the corresponding transcripts. We further reduce the size of this dataset to address the following challenges: First, with multiple annotations available from up to 11 different annotators per meeting, we need to keep only one annotation per meeting. Second, the meeting minutes can contain too many detailed items that are not suitable to be considered as topics. Third, the annotations do not necessarily point to a consecutive chunk of transcripts, but jump back and forth. To account for these issues, we keep only meetings with an annotation of at most 10 topics, and the annotations are not interspersed. With all the filters, we include 11 meetings into the TCR dataset. If there is no annotation for some parts of the transcripts, we follow the same logic described in Section 3.3.2 to fill the empty values. The original transcripts do not have timestamps, so we estimate the time information with the fixed rate of 150 words per minute.

### 3.4 Data Augmentation

All data sources described above provide ground truth topics for subsections of transcripts. However, the list of topics in the annotation only reflect the topic that are discussed. Real meetings do not always follow the planned agenda. Participants sometimes go off topic and have to skip some pre-arranged topics due to time limits. The TCR dataset schema is designed to test and evaluate such scenarios by incorporating the "variations" section. To reflect such scenarios, we also provide a script to either (1) add topics that are not discussed to a meeting as a planned topic or (2) remove topics and corresponding contents from a meeting. This can help enrich the TCR dataset to include a varied range of meeting styles. Implementation details can be found in the project repository.

For the augmented meetings, we keep the type of variation and the changed topic list in the "variations" field in the metadata for each meeting. Figure 2 shows an example of the change in metadata for an augmented meeting. If a topic is planned, but not discussed in a meeting, the topic is added to the "variation_addTopics" and the corresponding empty contents are also added to the "topic" section. Users can easily extend this by adding topics with non-empty contents to expand the simulation further. If we want to remove a topic and its corresponding contents together from a meeting, the changes are reflected in the "variation_removeTopics" list as well as the "topic" contents. The timestamps of the remaining contents are also changed accordingly. With this structure, we can test the relevance between the transcripts and topics that could have been planned but are not part of the actual conversation.

## 4 Topic-Conversation Relevance Benchmarks

We generate Topic-Conversation Relevance benchmarks on a selected subset[4] of the TCR dataset. Given the significant difference in meeting styles and structures, the benchmarks are reported for each data source separately.

### 4.1 Methodology

We use GPT-4 to create the benchmark results. For each meeting, we cut the transcripts into snippets with equal length based on timestamps. We conduct the experiments in duration length of 5 minutes, 10 minutes and 15 minutes. Then the prompt takes the snippet of transcripts and the full topic list from the meeting as inputs, and asks for an evaluation of the transcript's relevance to each topic in the list. The relevance score is represented by 4 levels: 0 means Not Relevant, 1 means Somewhat Relevant, 2 means Mostly Relevant, and 3 means Very Relevant. The detailed definitions of the relevance levels are given as a multiple-choice question in the prompt. In the development stage, we try different output requests, such as floats, integers, binaries and multiple choices. We present the final benchmark results all in the multiple choices style as it has been giving the most robust results across all data sources.

In the evaluation stage, we treat the Topic-Conversation Relevance problem as a binary classification. If based on the ground truth label, a topic is discussed for more than 30 seconds in the transcripts, then we mark it as "Discussed", otherwise "Not Discussed". For the GPT-4 responses, we treat "0 Not Relevant" as "Not Discussed", and everything else as "Discussed". In the results presented in Section4.2, we specifically focus on scenarios where the discussion is off-topic, so the "Not Discussed" topics are treated as positive cases. We use Precision ("LLM detects a topic is not being discussed and it is true") and Recall ("A topic is not being discussed and LLM detects it") as the main metrics. The full results treating "Discussed" and "Not Discussed" as positive cases respectively are shown in the Appendix A.3.

### 4.2 Results

The benchmark results focusing on the "Not Discussed" category are shown in Table 4. We split the results by data source and transcripts length.

---

[4]Selected subset: we select 30 random meetings from the MeetingBank dataset as the structures of meetings from this data sources are similar and can be represented by a subset; QMSum_AMI and QMSum_Parliament are not included in the benchmark because the former are mostly scenario discussions that are not real meetings and the style of latter is covered by MeetingBank and NCPC meetings.

Table 4: Topic-Conversation Relevance Benchmark Results

| Data Source | Transcripts Length | N Prompts | Prompt*Topic Pairs | F1 | Precision | Recall |
|---|---|---|---|---|---|---|
| SIM_syn100 | 5 min | 509 | 2,031 | 0.9587 | 0.9620 | 0.9554 |
|  | 10 min | 231 | 930 | 0.9272 | 0.8994 | 0.9568 |
|  | 15 min | 137 | 562 | 0.9175 | 0.8669 | 0.9744 |
| ICSI_original75 | 5 min | 790 | 5,175 | 0.8663 | 0.9615 | 0.7882 |
|  | 10 min | 382 | 2,502 | 0.8582 | 0.9462 | 0.7851 |
|  | 15 min | 244 | 1,594 | 0.8488 | 0.9243 | 0.7847 |
| ICSI_QMSum | 5 min | 550 | 3,079 | 0.7506 | 0.9373 | 0.6259 |
|  | 10 min | 266 | 1,492 | 0.7242 | 0.9441 | 0.5874 |
|  | 15 min | 170 | 955 | 0.7222 | 0.9381 | 0.5871 |
| MeetingBank _rnd30 | 5 min | 594 | 4,236 | 0.9804 | 0.9891 | 0.9720 |
|  | 10 min | 301 | 2,168 | 0.9767 | 0.9843 | 0.9691 |
|  | 15 min | 204 | 1,479 | 0.9688 | 0.9671 | 0.9705 |
| MeetingBank _ReAnnotated _rnd30 | 5 min | 594 | 4,236 | 0.9817 | 0.9913 | 0.9723 |
|  | 10 min | 301 | 2,168 | 0.9810 | 0.9895 | 0.9726 |
|  | 15 min | 204 | 1,479 | 0.9755 | 0.9824 | 0.9687 |
| NCPC | 5 min | 562 | 4,585 | 0.9702 | 0.9855 | 0.9553 |
|  | 10 min | 277 | 2,261 | 0.9631 | 0.9800 | 0.9468 |
|  | 15 min | 181 | 1,478 | 0.9614 | 0.9664 | 0.9565 |
| ELITR | 5 min | 70 | 584 | 0.8429 | 0.9390 | 0.7646 |
|  | 10 min | 31 | 261 | 0.8182 | 0.9184 | 0.7377 |
|  | 15 min | 20 | 166 | 0.8043 | 0.8706 | 0.7475 |

For the highly structured meetings (MeetingBank, NCPC), the benchmark results show very high precision and recall. Most of these meetings follow the pre-defined agenda topics strictly and often state the topic to-be-discussed at the beginning of the section. Different annotations do not impact the results much. The other less structured meetings, such as project meetings (ICSI, ELITR) and brainstorming discussions (SIM), are more challenging. Most of these meetings do not have clear statements separating different topics and related sub-topics are often discussed back and forth. Different topic annotations can impact the results significantly.

We also notice that if there are multiple topics included in the same snippet of transcripts (8), it is even more challenging to correctly predict the relevance comparing with single-topic transcript (9). This could be due to the fact that transitions between topics are not always clear in the less structured meetings. Results split by topic counts are also included in Appendix A.3.

## 5 Future Work

The dataset can be further improved by including more types of meetings in different domains. However, it is particularly hard to obtain real day-to-day meetings in a working environment as most of such meetings consist sensitive business information. Hence the project team is working on inviting domain experts (e.g., legal, healthcare, finance, etc.) to create meeting agendas for different types of meetings in their industry, and conducting domain-specific meetings based on the agendas. We are currently in the data collection stage using the same method as the SIM dataset, with additional requirements on participants' professional experience.

In addition to English, we are also working on integrating other languages into the dataset. One of the efforts is to translate the current data sources into other languages with reliable translation services and test the performance on the same tasks.

A challenge of evaluating topic-conversation relevance is the blurred boundaries between topics. At a meeting structure level, a certain chunk of transcripts can be marked as belonging to a topic, but it is very likely that some parts of the conversation are actually not directly related to the topic or

even belong to another listed topic. It would be desirable to create sub-labels at sentence or group of sentences level to capture relevance scores at a lower granularity.

Additionally, we believe it would be beneficial to include audio data in the TCR dataset along with transcripts. We will work on aggregating audio data for multiple data sources (SIM data and other public data) into the dataset.

## Acknowledgments and Disclosure of Funding

We acknowledge Naglakshmi Chande, Quchen Fu, Xavier Gitiaux and Thierry Tremblay from the Microsoft Teams team for all the brainstorming, analysis reviews and infrastructure work. We acknowledge the data donation workers for the SIM dataset who were compensated for their time and effort. No competing interests are associated with this work.

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

# A  Appendix

## A.1  Exploratory Analysis per Meeting by Data Source

Table 5: Exploratory Analysis per Meeting by Data Source

| Data Source | Metric | Meeting Duration (minutes) | N Speakers per Meeting | N Topics per Meeting | Topic Duration (minutes) | Topic Text Lenth (words) |
|---|---|---|---|---|---|---|
| SIM | mean | 34.24 | 4.00 | 1.00 | 34.24 | 11.32 |
| | std | 5.47 | 0.00 | 0.00 | 5.47 | 4.43 |
| SIM_syn100 | mean | 27.24 | 4.00 | 3.48 | 7.83 | 10.72 |
| | std | 9.12 | 0.00 | 1.11 | 1.90 | 4.63 |
| ICSI | mean | 45.19 | 6.20 | 6.52 | 6.93 | 2.85 |
| | std | 14.61 | 1.35 | 2.41 | 9.41 | 1.97 |
| QMSum_ICSI | mean | 44.88 | 6.31 | 4.27 | 10.40 | 4.15 |
| | std | 12.08 | 1.34 | 1.09 | 6.79 | 2.32 |
| QMSum_AMI | mean | 34.03 | 4.00 | 3.94 | 8.45 | 6.76 |
| | std | 12.87 | 0.00 | 0.93 | 6.27 | 3.01 |
| QMSum_Parliament | mean | 92.21 | 23.80 | 6.45 | 13.90 | 8.43 |
| | std | 29.68 | 23.81 | 1.43 | 14.13 | 4.85 |
| MeetingBank | mean | 109.86 | 8.54 | 5.98 | 18.36 | 59.47 |
| | std | 90.91 | 3.20 | 3.92 | 33.18 | 35.41 |
| MeetingBank _rnd30 | mean | 95.31 | 8.33 | 6.30 | 15.13 | 58.93 |
| | std | 70.50 | 2.72 | 4.09 | 24.15 | 38.32 |
| MeetingBank _ReAnnotated | mean | 109.86 | 8.54 | 5.97 | 18.39 | 10.03 |
| | std | 90.91 | 3.20 | 3.91 | 33.20 | 5.63 |
| MeetingBank _ReAnnotated_rnd30 | mean | 95.31 | 8.33 | 6.30 | 15.13 | 9.69 |
| | std | 70.50 | 2.72 | 4.09 | 24.15 | 2.55 |
| NCPC | mean | 141.69 | 25.60 | 8.00 | 17.71 | 6.27 |
| | std | 64.29 | 7.74 | 2.05 | 26.48 | 3.68 |
| ELITR | mean | 34.26 | 5.45 | 7.64 | 4.01 | 6.95 |
| | std | 34.95 | 2.38 | 2.01 | 6.68 | 4.76 |

## A.2  SIM Dataset Unique Speaker Metadata

Table 6: SIM Dataset - Unique Speaker Metadata by Age and Gender

| Age | Gender | | Total |
|---|---|---|---|
| | Female | Male | |
| 18-24 | 12 | 8 | 20 |
| 25-34 | 28 | 22 | 50 |
| 35-44 | 18 | 14 | 32 |
| 45+ | 24 | 23 | 47 |
| Total | 82 | 67 | 149 |

## A.3  Complete Evaluation Results

Complete benchmark results by positive classes, topic counts and snippet sizes are reported in Table 7 to Table 12.

Table 7: Benchmark Results - Positive Class: Not Discussed - All Snippets

| Data Source | Transcripts Length | N Prompts | Prompt*Topic Pairs | F1 | Precision | Recall |
|---|---|---|---|---|---|---|
| SIM_syn100 | 5 min | 509 | 2,031 | 0.9587 | 0.9620 | 0.9554 |
| | 10 min | 231 | 930 | 0.9272 | 0.8994 | 0.9568 |
| | 15 min | 137 | 562 | 0.9175 | 0.8669 | 0.9744 |
| ICSI_original75 | 5 min | 790 | 5,175 | 0.8663 | 0.9615 | 0.7882 |
| | 10 min | 382 | 2,502 | 0.8582 | 0.9462 | 0.7851 |
| | 15 min | 244 | 1,594 | 0.8488 | 0.9243 | 0.7847 |
| ICSI_QMSum | 5 min | 550 | 3,079 | 0.7506 | 0.9373 | 0.6259 |
| | 10 min | 266 | 1,492 | 0.7242 | 0.9441 | 0.5874 |
| | 15 min | 170 | 955 | 0.7222 | 0.9381 | 0.5871 |
| MeetingBank _rnd30 | 5 min | 594 | 4,236 | 0.9804 | 0.9891 | 0.9720 |
| | 10 min | 301 | 2,168 | 0.9767 | 0.9843 | 0.9691 |
| | 15 min | 204 | 1,479 | 0.9688 | 0.9671 | 0.9705 |
| MeetingBank _ReAnnotated _rnd30 | 5 min | 594 | 4,236 | 0.9817 | 0.9913 | 0.9723 |
| | 10 min | 301 | 2,168 | 0.9810 | 0.9895 | 0.9726 |
| | 15 min | 204 | 1,479 | 0.9755 | 0.9824 | 0.9687 |
| NCPC | 5 min | 562 | 4,585 | 0.9702 | 0.9855 | 0.9553 |
| | 10 min | 277 | 2,261 | 0.9631 | 0.9800 | 0.9468 |
| | 15 min | 181 | 1,478 | 0.9614 | 0.9664 | 0.9565 |
| ELITR | 5 min | 70 | 584 | 0.8429 | 0.9390 | 0.7646 |
| | 10 min | 31 | 261 | 0.8182 | 0.9184 | 0.7377 |
| | 15 min | 20 | 166 | 0.8043 | 0.8706 | 0.7475 |

Table 8: Benchmark Results - Positive Class: Not Discussed - Snippets with Multiple Topics

| Data Source | Transcripts Length | N Prompts | Prompt*Topic Pairs | F1 | Precision | Recall |
|---|---|---|---|---|---|---|
| SIM_syn100 | 5 min | 263 | 1,050 | 0.9312 | 0.9183 | 0.9445 |
| | 10 min | 213 | 861 | 0.9202 | 0.8887 | 0.9540 |
| | 15 min | 137 | 562 | 0.9175 | 0.8669 | 0.9744 |
| ICSI_original75 | 5 min | 333 | 2,182 | 0.8302 | 0.9403 | 0.7432 |
| | 10 min | 253 | 1,659 | 0.8364 | 0.9284 | 0.7609 |
| | 15 min | 202 | 1,324 | 0.8360 | 0.9129 | 0.7710 |
| ICSI_QMSum | 5 min | 195 | 1,092 | 0.7379 | 0.9167 | 0.6175 |
| | 10 min | 170 | 957 | 0.7191 | 0.9320 | 0.5854 |
| | 15 min | 140 | 788 | 0.7345 | 0.9308 | 0.6066 |
| MeetingBank _rnd30 | 5 min | 140 | 1,005 | 0.9663 | 0.9682 | 0.9645 |
| | 10 min | 114 | 825 | 0.9593 | 0.9656 | 0.9531 |
| | 15 min | 96 | 699 | 0.9444 | 0.9297 | 0.9597 |
| MeetingBank _ReAnnotated _rnd30 | 5 min | 140 | 1,005 | 0.9588 | 0.9726 | 0.9454 |
| | 10 min | 114 | 825 | 0.9624 | 0.9735 | 0.9515 |
| | 15 min | 96 | 699 | 0.9533 | 0.9592 | 0.9476 |
| NCPC | 5 min | 81 | 668 | 0.9320 | 0.9621 | 0.9038 |
| | 10 min | 67 | 555 | 0.9121 | 0.9316 | 0.8934 |
| | 15 min | 66 | 539 | 0.9341 | 0.9174 | 0.9514 |
| ELITR | 5 min | 41 | 343 | 0.8258 | 0.9143 | 0.7529 |
| | 10 min | 21 | 181 | 0.8057 | 0.8763 | 0.7456 |
| | 15 min | 16 | 138 | 0.8056 | 0.8406 | 0.7733 |

Table 9: Benchmark Results - Positive Class: Not Discussed - Snippets with Only One Topic

| Data Source | Transcripts Length | N Prompts | Prompt*Topic Pairs | F1 | Precision | Recall |
|---|---|---|---|---|---|---|
| SIM_syn100 | 5 min | 246 | 981 | 0.9818 | 1.0000 | 0.9643 |
| | 10 min | 18 | 69 | 0.9901 | 1.0000 | 0.9804 |
| | 15 min | 0 | 0 | - | - | - |
| ICSI_original75 | 5 min | 457 | 2,993 | 0.8888 | 0.9743 | 0.8172 |
| | 10 min | 129 | 843 | 0.8934 | 0.9745 | 0.8247 |
| | 15 min | 42 | 270 | 0.8988 | 0.9681 | 0.8387 |
| ICSI_QMSum | 5 min | 355 | 1,987 | 0.7566 | 0.9471 | 0.6299 |
| | 10 min | 96 | 535 | 0.7317 | 0.9623 | 0.5903 |
| | 15 min | 30 | 167 | 0.6733 | 0.9714 | 0.5152 |
| MeetingBank _rnd30 | 5 min | 454 | 3,231 | 0.9845 | 0.9952 | 0.9741 |
| | 10 min | 187 | 1,343 | 0.9862 | 0.9946 | 0.9779 |
| | 15 min | 108 | 780 | 0.9877 | 0.9969 | 0.9786 |
| MeetingBank _ReAnnotated _rnd30 | 5 min | 454 | 3,231 | 0.9882 | 0.9967 | 0.9800 |
| | 10 min | 187 | 1,343 | 0.9911 | 0.9982 | 0.9841 |
| | 15 min | 108 | 780 | 0.9923 | 1.0000 | 0.9847 |
| NCPC | 5 min | 481 | 3,917 | 0.9754 | 0.9886 | 0.9625 |
| | 10 min | 210 | 1,706 | 0.9755 | 0.9917 | 0.9599 |
| | 15 min | 115 | 939 | 0.9735 | 0.9887 | 0.9587 |
| ELITR | 5 min | 29 | 241 | 0.8640 | 0.9701 | 0.7788 |
| | 10 min | 10 | 80 | 0.8403 | 1.0000 | 0.7246 |
| | 15 min | 4 | 28 | 0.8000 | 1.0000 | 0.6667 |

Table 10: Benchmark Results - Positive Class: Discussed - All Snippets

| Data Source | Transcripts Length | N Prompts | Prompt*Topic Pairs | F1 | Precision | Recall |
|---|---|---|---|---|---|---|
| SIM_syn100 | 5 min | 509 | 2,031 | 0.9234 | 0.9176 | 0.9293 |
| | 10 min | 231 | 930 | 0.9148 | 0.9492 | 0.8829 |
| | 15 min | 137 | 562 | 0.9346 | 0.9799 | 0.8933 |
| ICSI_original75 | 5 min | 790 | 5,175 | 0.6494 | 0.5156 | 0.8771 |
| | 10 min | 382 | 2,502 | 0.7093 | 0.5957 | 0.8765 |
| | 15 min | 244 | 1,594 | 0.7508 | 0.6618 | 0.8676 |
| ICSI_QMSum | 5 min | 550 | 3,079 | 0.5439 | 0.3987 | 0.8555 |
| | 10 min | 266 | 1,492 | 0.6216 | 0.4711 | 0.9136 |
| | 15 min | 170 | 955 | 0.6896 | 0.5485 | 0.9284 |
| MeetingBank _rnd30 | 5 min | 594 | 4,236 | 0.9065 | 0.8702 | 0.9459 |
| | 10 min | 301 | 2,168 | 0.9061 | 0.8787 | 0.9354 |
| | 15 min | 204 | 1,479 | 0.8896 | 0.8951 | 0.8841 |
| MeetingBank _ReAnnotated _rnd30 | 5 min | 594 | 4,236 | 0.9130 | 0.8727 | 0.9573 |
| | 10 min | 301 | 2,168 | 0.9238 | 0.8929 | 0.9569 |
| | 15 min | 204 | 1,479 | 0.9167 | 0.8953 | 0.9390 |
| NCPC | 5 min | 562 | 4,585 | 0.8427 | 0.7788 | 0.9180 |
| | 10 min | 277 | 2,261 | 0.8432 | 0.7857 | 0.9098 |
| | 15 min | 181 | 1,478 | 0.8553 | 0.8391 | 0.8721 |
| ELITR | 5 min | 70 | 584 | 0.5976 | 0.4734 | 0.8099 |
| | 10 min | 31 | 261 | 0.6875 | 0.5789 | 0.8462 |
| | 15 min | 20 | 166 | 0.7568 | 0.6914 | 0.8358 |

Table 11: Benchmark Results - Positive Class: Discussed - Snippets with Multiple Topics

| Data Source | Transcripts Length | N Prompts | Prompt*Topic Pairs | F1 | Precision | Recall |
|---|---|---|---|---|---|---|
| SIM_syn100 | 5 min | 263 | 1,050 | 0.9071 | 0.9247 | 0.8901 |
| | 10 min | 213 | 861 | 0.9122 | 0.9492 | 0.8779 |
| | 15 min | 137 | 562 | 0.9346 | 0.9799 | 0.8933 |
| ICSI_original75 | 5 min | 333 | 2,182 | 0.6685 | 0.5441 | 0.8667 |
| | 10 min | 253 | 1,659 | 0.7258 | 0.6224 | 0.8704 |
| | 15 min | 202 | 1,324 | 0.7577 | 0.6739 | 0.8654 |
| ICSI_QMSum | 5 min | 195 | 1,092 | 0.6275 | 0.4913 | 0.8681 |
| | 10 min | 170 | 957 | 0.6734 | 0.5321 | 0.9169 |
| | 15 min | 140 | 788 | 0.7221 | 0.5915 | 0.9267 |
| MeetingBank _rnd30 | 5 min | 140 | 1,005 | 0.8787 | 0.8727 | 0.8848 |
| | 10 min | 114 | 825 | 0.8815 | 0.8651 | 0.8986 |
| | 15 min | 96 | 699 | 0.8564 | 0.8930 | 0.8227 |
| MeetingBank _ReAnnotated _rnd30 | 5 min | 140 | 1,005 | 0.8596 | 0.8201 | 0.9032 |
| | 10 min | 114 | 825 | 0.8925 | 0.8643 | 0.9227 |
| | 15 min | 96 | 699 | 0.8884 | 0.8756 | 0.9015 |
| NCPC | 5 min | 81 | 668 | 0.8460 | 0.7900 | 0.9105 |
| | 10 min | 67 | 555 | 0.8397 | 0.8088 | 0.8730 |
| | 15 min | 66 | 539 | 0.8712 | 0.9034 | 0.8413 |
| ELITR | 5 min | 41 | 343 | 0.6335 | 0.5263 | 0.7955 |
| | 10 min | 21 | 181 | 0.7285 | 0.6548 | 0.8209 |
| | 15 min | 16 | 138 | 0.7879 | 0.7536 | 0.8254 |

Table 12: Benchmark Results - Positive Class: Discussed - Snippets with Only One Topic

| Data Source | Transcripts Length | N Prompts | Prompt*Topic Pairs | F1 | Precision | Recall |
|---|---|---|---|---|---|---|
| SIM_syn100 | 5 min | 246 | 981 | 0.9509 | 0.9065 | 1.0000 |
| | 10 min | 18 | 69 | 0.9730 | 0.9474 | 1.0000 |
| | 15 min | 0 | 0 | - | - | - |
| ICSI_original75 | 5 min | 457 | 2,993 | 0.6290 | 0.4865 | 0.8893 |
| | 10 min | 129 | 843 | 0.6584 | 0.5197 | 0.8980 |
| | 15 min | 42 | 270 | 0.6963 | 0.5732 | 0.8868 |
| ICSI_QMSum | 5 min | 355 | 1,987 | 0.4847 | 0.3399 | 0.8443 |
| | 10 min | 96 | 535 | 0.4987 | 0.3444 | 0.9029 |
| | 15 min | 30 | 167 | 0.5000 | 0.3402 | 0.9429 |
| MeetingBank _rnd30 | 5 min | 454 | 3,231 | 0.9183 | 0.8692 | 0.9732 |
| | 10 min | 187 | 1,343 | 0.9297 | 0.8913 | 0.9716 |
| | 15 min | 108 | 780 | 0.9389 | 0.8978 | 0.9840 |
| MeetingBank _ReAnnotated _rnd30 | 5 min | 454 | 3,231 | 0.9370 | 0.8964 | 0.9814 |
| | 10 min | 187 | 1,343 | 0.9543 | 0.9207 | 0.9905 |
| | 15 min | 108 | 780 | 0.9615 | 0.9259 | 1.0000 |
| NCPC | 5 min | 481 | 3,917 | 0.8414 | 0.7745 | 0.9210 |
| | 10 min | 210 | 1,706 | 0.8462 | 0.7674 | 0.9429 |
| | 15 min | 115 | 939 | 0.8327 | 0.7589 | 0.9224 |
| ELITR | 5 min | 29 | 241 | 0.5234 | 0.3784 | 0.8485 |
| | 10 min | 10 | 80 | 0.5366 | 0.3667 | 1.0000 |
| | 15 min | 4 | 28 | 0.5000 | 0.3333 | 1.0000 |

