# Supplementary Materials

## Topic-Conversation Relevance (TCR) Dataset and Benchmarks

## 1. Data and Repository Access

- For NeurIPS review purpose, please download the repo and data by the **link here**. Included in the download:
  - Data in `\data` folder
  - Scripts to augment data, create synthetic meetings and help reproduce benchmark results
  - README, LICENSE, LICENSE_Data and other related documents
- The repo with same contents will be made public later:
  - https://github.com/microsoft/topic_conversation
  - Metadata link

## 2. License

Please refer to the LICENSE and LICENSE_Data file in the repo.  To access the repo contents, please refer to Section 1. Data and Repository Access.

## 3. Dataset Documentation

Following styles suggested in Datasheets for Datasets.

### Motivation

*For what purpose was the dataset created?*
The TCR dataset is created to help research about meeting and conversation contents. In the *Topic-Conversation Relevance (TCR) Dataset and Benchmarks* paper, we focus on studying the relevance of a particular topic with a conversation. We build benchmarks about topic-conversation relevance with the TCR dataset.

*Who created the dataset?*
The dataset was created by Microsoft.

*Any other comments?*
None.

### Composition

*What do the instances that comprise the dataset represent? Please provide a description.*
The dataset contains meeting transcripts and their corresponding topics. For each meeting and topic, metadata about start and end timestamps, lines and total word counts are available.

*How many instances are there in total?*
In total there are 1,506 unique meetings, over 22,000,000 words in transcripts, and over 15,000 topic annotations. Details can be found in the *Topic-Conversation Relevance (TCR) Dataset and Benchmarks* paper.

*Does the dataset contain all possible instances or is it a sample (not necessarily random) of instances from a larger set?*
The dataset is a sample of possible meeting instances. It is designed to cover a variety of domains and meeting styles of the total population.

*What data does each instance consist of?*
Each instance (meeting) has pieces of information: "metadata" and "topics". The "metadata" provides meeting-specific source, time and word count information. It also has a "variations" field for potential data augmentation. The "topics" contains all topics and their corresponding information and transcripts.

*Is there a label or target associated with each instance?*
The topic titles are the topic label for the corresponding transcript section.

*Is any information missing from individual instances?*
No, all fields are provided for each instance.

*Are relationships between individual instances made explicit?*
Each individual instance (meeting) has its unique name. Meetings are independent and do not have relationships between each other.

*Are there recommended data splits?*
No, the data is not split. The benchmark is built to test a selected subset of instances, not training. The user can easily do the split for non-testing purposes.

*Are there any errors, sources of noise, or redundancies in the dataset?*
The transcripts may have transcribing errors. The topic annotations may not be exactly accurate due to annotator errors.

*Is the dataset self-contained, or does it link to or otherwise rely on external resources?*
Yes, the dataset is self-contained.

*Does the dataset contain data that might be considered confidential?*
No.

*Does the dataset contain data that, if viewed directly, might be offensive, insulting, threatening, or might otherwise cause anxiety?*
There may be contents that meeting participants express their own opinions in their personal style. The contents do not convey the intents of Microsoft nor the authors.

*Does the dataset identify any subpopulations (e.g., by age, gender*
No.

*Is it possible to identify individuals, either directly or indirectly from the dataset?*
Not possible directly from the TCR dataset. If the original data sources have metadata with individual information, based on the transcript's contents, the metadata can be retrieved. For the SIM data created by the authors, it is not possible to identify individuals.

*Does the dataset contain data that might be considered sensitive in any way?*
There may be contents that meeting participants express their own opinions in their personal style. The contents do not convey the intents of Microsoft nor the authors.

*Any other comments?*
None.

Collection Process

*How was the data associated with each instance acquired?*
The public datasets are collected from the data sources. The SIM dataset is collected by the project team. For each meeting, 4 participants (consent acquired) join an online meeting to discuss a topic and actively interact with each other. Meeting contents are recorded, and transcripts are generated.

*What mechanisms or procedures were used to collect the data?*
Microsoft Teams for the SIM data collection.

*If the dataset is a sample from a larger set, what was the sampling strategy?*
No explicit sampling procedure. We try to keep all meetings with pre-meeting style topics from all input data sources. Refer to Section 3 in the *Topic-Conversation Relevance (TCR) Dataset and Benchmarks* paper.

*Who was involved in the data collection process and how were they compensated?*
For the SIM data collection, we worked with a third-party data collection company and the company was compensated by contract.

*Over what timeframe was the data collected?*
For the SIM data collection, the effort was from 2021-2024. For the public data aggregation, the effort was done in 2024.

*Were any ethical review processes conducted?*
For the SIM data collection, internal privacy and legal reviews.

*Did you collect the data from the individuals in question directly, or obtain it via third parties or other sources? Were the individuals in question notified about the data collection? Did the individuals in question consent to the collection and use of their data?*
For the SIM data collection, consent forms are acquired from each individual participant. For the other public datasets, we collect the data directly from the source, ensuring to respect the licenses of the data.

*If consent was obtained, were the consenting individuals provided with a mechanism to revoke their consent in the future or for certain uses?*
No.

*Has an analysis of the potential impact of the dataset and its use on data subjects (e.g., a data protection impact analysis) been conducted?*
No

*Any other comments?*
None.

## Preprocessing, Cleaning, Labeling

*Was any preprocessing/cleaning/labeling of the data done?*
Yes. We keep only meeting with certain length and topic annotation properties. We process transcripts and re-annotated topics for certain data sources. For details, please refer to Section 3 in the *Topic-Conversation Relevance (TCR) Dataset and Benchmarks* paper.

*Was the "raw" data saved in addition to the preprocessed/cleaned/labeled data?*
For the SIM data, everything is included. For the other data sources, raw data can be obtained from the original data source. For meetings with re-annotated topics, both sets are kept in the TCR data set.

*Is the software that was used to preprocess/clean/label the data available? If so, please provide a link or other access point.*
No.

*Any other comments?*
None.

## Uses

*Has the dataset been used for any tasks already?*
We build benchmarks about topic-conversation relevance with the TCR dataset.

*Is there a repository that links to any or all papers or systems that use the dataset?*
The repo will be public later: https://github.com/microsoft/topic_conversation.

*What (other) tasks could the dataset be used for?*
The TCR dataset can be used for any NLP, transcripts, or meeting related analysis.

*Is there anything about the composition of the dataset or the way it was collected and preprocessed/cleaned/labeled that might impact future uses?*
Not aware of any.

*Are there tasks for which the dataset should not be used?*
N/A

*Any other comments?*
None.

*Will the dataset be distributed to third parties outside of the entity on behalf of which the dataset was created?*
Yes, it will be available under the specified LICENSE in the project repo.

*How will the dataset will be distributed?*
Through the project GitHub repo.

*Does the dataset have a digital object identifier (DOI)?*
Not at the moment. To be created when the data is made public.

*Will the dataset be distributed under a copyright or other intellectual property (IP) license, and/or under applicable terms of use (ToU)?*
Please refer to the project repo

*Have any third parties imposed IP-based or other restrictions on the data associated with the instances?*
No

*Do any export controls or other regulatory restrictions apply to the dataset or to individual instances?*
No

*Any other comments?*
None.

Maintenance

*Who will be supporting/hosting/maintaining the dataset?*
The authors and team.

*How can the owner/curator/manager of the dataset be contacted?*
Please refer to the *Topic-Conversation Relevance (TCR) Dataset and Benchmarks* paper or the project GitHub repo for author contacts.

*Is there an erratum?*
No.

*Will the dataset be updated?*
We will update for correcting potential errors or including new instance.

*If the dataset relates to people, are there applicable limits on the retention of the data associated with the instances?*
N/A

*Will older versions of the dataset continue to be supported/hosted/maintained?*
Yes. We will be maintaining the repo and data links.

*If others want to extend/augment/build on/contribute to the dataset, is there a mechanism for them to do so?*
Please refer to the project repo.

*Any other comments?*
None.

# 4. Author Statement

The authors bear all responsibility in case of violation of rights.