# OpenReview forum: "Topic-Conversation Relevance (TCR)  Dataset and Benchmarks"
_NeurIPS.cc/2024/Datasets_and_Benchmarks_Track — NeurIPS 2024 Track Datasets and Benchmarks Poster_

### Official Review · Reviewer_8mfY · 2024-07-11
**Reviews**

**Rating:** 4
**Confidence:** 3
**Correctness:** N.A
**Clarity:** N.A

**Review:**

Strength:
1. The paper has a well-defined structure that logically progresses from the introduction of the TCR Dataset to the detailed analysis and discussion of the results.
2. The TCR Dataset is extensive, covering a wide range of meeting styles and domains, which provides a rich resource for researchers and developers.

Weakness:
1. The paper does not clearly articulate the significance of the dataset, particularly in the context of meetings that are intended for brainstorming sessions where the agenda may be fluid and the discussions are meant to be exploratory rather than strictly adhering to predefined topics.
2. The document does not sufficiently address whether the extensive corpus of data is genuinely required. Given the impressive capabilities of GPT-4 without extensive training, it raises questions about the necessity of such a large dataset for the task at hand.
3. The effectiveness of GPT-4 even without training suggests that the dataset might be overreaching. With the right prompting, GPT-4 could potentially achieve even better results.
4. The abstract said “benchmarks are created using GPT-4 to evaluate the model accuracy”, what’s the “model” here? It seems only GPT-4 is used in the paper.

**Strengths:**

1. The paper has a well-defined structure that logically progresses from the introduction of the TCR Dataset to the detailed analysis and discussion of the results.
2. The TCR Dataset is extensive, covering a wide range of meeting styles and domains, which provides a rich resource for researchers and developers.

**Additional Feedback:**

N.A

**Documentation:**

N.A

**Ethics:**

N.A

**Limitations:**

N.A

**Opportunities For Improvement:**

See weaknesses

**Relation To Prior Work:**

N.A

**Summary And Contributions:**

The document introduces the Topic-Conversation Relevance (TCR) Dataset, a comprehensive collection of meeting transcripts and topics designed to assess the relevance of conversation to predefined agendas.
The TCR dataset aims to improve meeting effectiveness by providing a benchmark for evaluating the focus of discussions. The paper also discusses the application of GPT-4 in generating synthetic meetings and evaluating model accuracy in understanding transcription-topic relevance. The paper concludes with future work directions, including expanding the dataset with more meeting types and improving topic annotations.

---

> ### Author Rebuttal · Authors · 2024-08-16
>
> Dear Reviewer,
>
> We sincerely appreciate your time and feedback. In this response, we are providing additional details regarding your concerns and suggestions. We hope our responses can help address your comments.
>
> 1.	Regarding brainstorming-like meetings: Thank you for bringing this up. This is a very good point, and we are providing some explanations and mitigation proposal below. We can discuss this from 3 perspectives: data schema, benchmark method and potential production goals.
>     * First, from the data schema point of view, yes, currently each line of transcript belongs to one topic and one topic covers a consecutive section. However, the schema is flexible so that in a meeting where people discuss topics back and forth (A1, B1, A2, B2), topic A can still contain transcripts from A1 and A2, while topic B contains transcripts B1 and B2. The line numbers and transcript timestamp can help users get the correct chronological order if desired.
>     * Second, the benchmark method that we present can pick multiple topics being discussed in the given snippet of transcript. This is because the benchmark prompt asks for a 0-3 evaluation for each possible topic of the meeting, instead of asking which one is being discussed. Hence, the benchmark method can evaluate any arbitrary number of candidate topics independently.
>     * Lastly, one of the most important production goals is to help people stay on track of the agenda topics defined ahead of the meetings. So, it is critical to know which type of meeting can benefit the most from such reminders. We did a meeting taxonomy study as part of the project and are planning to build such a classifier to predict meeting types based on the meeting invite, agenda contents and first X minutes of transcripts. Hence, if a meeting is predicted to be a brainstorming session, the topic moderator should either be disabled or only tracking the high-level topic. The latter is exactly the test results on the SIM (original) dataset. We only presented the results for SIM_syn100 in the paper to demonstrate the multi-topic scenario. Thank you for raising questions regarding the single-topic brainstorming type of meetings. We will include the results for SIM (original, single-topic brainstorming meetings) in the paper in the revision.
> 2.	Regarding training: We completely agree that several existing LLM models are sophisticated enough and do not need extensive training. Hence, the goal of the dataset is not to help train such models further, but to help evaluate and benchmark the prompts or model outputs.    Such a dataset is required to answer, “what’s the performance of a LLM model on such a task”. It will help researchers to do prompt engineering to achieve acceptable performance. Without such a test dataset this would be impossible.
>     * For example, we have been developing a feature based on the Topic-Conversation Relevance task results internally for several months. During this time, multiple versions of GPT-4 models were released. To switch a model to a newer version, we need to test the new model and prompts against the existing benchmarks, to make sure that the performance is stable. In one of the versions, we did notice some systematic regression on a certain data source compared with the benchmark. So, it led to more investigations and prompt engineering efforts on the new version.
>
>     We hope our dataset can help researchers build better benchmarks and test sets. We try to include a wide range of meeting styles, domains and topic styles in the TCR dataset to help achieve this goal.
>
> 3.	Regarding prompt engineering: We completely agree that with better prompt engineering, the LLM can achieve satisfactory results. That’s indeed what we did in the process of creating the benchmarks. The current benchmark results are generated from a robust prompt after many iterations of prompt engineering and tests. A very brief description of our workflow is:
>     * (1) Create the TCR dataset
>     * (2) Create a prompt to solve the TCR task with GPT-4
>     * (3) Evaluate the performance on the TCR dataset.
>     * (4) Prompt engineering by iterating through step 2 and 3
>     * (5) Report the benchmark results shown in the paper
>
>     With such a process, we developed better prompts and improved our benchmark recall by 4%~10% while maintaining the same precision.
>
>     There  is no training or fine-tuning involved in the process. We would like to provide the results generated from the original GPT-4 model as a baseline, so that researchers can improve the task performance with better models or better prompts in the future.
>
> 4.	Regarding GPT-4: Yes, we only use the GPT-4 (specifically GPT-4 -32k) model to generate benchmark results. It was the most widely used and stable version when we started the project. We will make the reference clearer in the revision.
>
> Thank you once again for your feedback and suggestions. If you have any other input, we are more than happy to address them accordingly.

---

### Official Review · Reviewer_CrsX · 2024-07-22
**A novel dataset and benchmark**

**Rating:** 7
**Confidence:** 4
**Correctness:** Yes
**Clarity:** Yes

**Review:**

The proposed dataset are novel and interesting. It would be better if the authors could manually evaluate the quality of the re-annotate topics generated by GPT-4 prompts.

**Strengths:**

N/A

**Additional Feedback:**

N/A

**Documentation:**

Yes

**Opportunities For Improvement:**

N/A

**Relation To Prior Work:**

Yes

**Summary And Contributions:**

This paper proposed a topic conversation relevance dataset, which contains large-scale different types of conversations.

---

> ### Author Rebuttal · Authors · 2024-08-16
>
> Dear Reviewer,
>
> We sincerely appreciate your time and feedback. In this response, we are providing additional details regarding your concerns and suggestions. Hope our responses can help address your comments.
>
> 1.	Regarding evaluation: We indeed evaluated a subset of the re-annotated topics. We selected 100 samples from all MeetingBank topics, and 3 team members provided their ratings on each sample respectively. The score range is 1-5. Across the 300 votes, the mean quality score was 4.6 and the median was 5. Thank you so much for pointing this out. We will add this information to the paper.
>
> Thank you once again for your feedback and suggestions. If you have any other input, we are more than happy to address them accordingly.

---

### Official Review · Reviewer_aWey · 2024-07-24
**Review of a dataset of meeting transcripts and a benchmark**

**Rating:** 8
**Confidence:** 3
**Clarity:** Yes

**Review:**

This is an interesting dataset and a strong benchmark. These are also the pros of this study while I only see two cons:

### Cons
* An ordinal regression task is casted as binary classification (solvable)
* The exploratory analysis is very poor.

**Strengths:**

Discussed in the review.

**Additional Feedback:**

* The repo is said not to be public: when will it be public?
* L120: How are the 5' defined?
* L123: Contents or words?
* L251: How did this time-based rule emerge? Why not 1' and how can it be the same time range across meetings (not all are lengthy)?
* L265 (more challenging): Particularly in Recall, right? Perhaps worth clarifying.
* L265 (Different ...): Where is this shown exactly? Please elaborate more in the text.
* L267 (evaluate the relevance): Confusing wording IMO; consider using predict/detect for the prediction stage and evaluation for the assessment of those predictions.
* L274 (from the same industry): How is this solving the problem discussed above?
* L277 (may not be...): The language could be improved. Also, what is an ideal pre-meeting way?
* L266: Please consider adding an example.
* L288: source > sources

**Correctness:**

* The abstract is misleading: 22m words not 22k.

**Documentation:**

Yes

**Ethics:**

There may be anonymisation issues - can't tell.

**Opportunities For Improvement:**

Discussed in the review, some suggestions follow:

* L250: This is an ordinal regression task. It is fine to report classification results but you should and also show regression metrics (MAE, MSE).
* L251: How did this time-based rule emerge? Why not 1' and how can it be the same time range across meetings (not all are lengthy)?
* L261: Please add F1 in the Table and consider ranking the sources based on this metric, averaged across the three source transcript lengths. Also, please add MAE and/or MSE to complement the results.

**Relation To Prior Work:**

Yes

**Summary And Contributions:**

This paper presents a dataset of meeting transcripts, time-stamped (minutes) and annotated regarding the relevance to pre-defined topics. GPT4 is used to classify topic relevance showing that some sources are easier to monitor (regarding relevance) compared to others.

---

> ### Author Rebuttal · Authors · 2024-08-16
>
> Dear Reviewer,
>
> We sincerely appreciate your time and feedback. In this response, we are providing additional details regarding your concerns and suggestions. Hope our responses can help address your comments.
>
> 1.	Regarding regression task: Thank you for raising this. We’d like to first explain why we decided to go with classification and then discuss how to address your concerns. The reason that we decided to use classification is that the task is given in a multiple-choice question style. Though it is marked as 0, 1, 2, 3, but it should be treated as A, B, C, D as in a multi-choice question. Because it is hard to explain the scores quantitatively. For example, the difference between 0 (Not Relevant) and 1 (Somewhat Relevant) can be much larger than the difference between 2 (Mostly Relevant) and 3 (Very Relevant); but numerically the delta is both 1. During our investigations, we actually did some analysis treating them as numbers and calculated the correlations against the ground truth. We did not report those numbers as we focused more on the binary evaluations in the later stage. To address your concerns, we propose to add the analysis of correlations and MSE into  the Appendix for reference.
> 2.	Regarding exploratory analysis: This is a great suggestion. We will add more exploratory analysis in the revision. We should include analysis of (but not limited to): meeting length, meeting size, topic counts, topic word counts, average length per topic, etc. We would also show data source breakdowns in the Appendix to help researchers select the ones that meet their requirements the most. Thank you for pointing this out.
> 3.	Regarding time-based rules:
>     * L251 (30s): This involves some subjective evaluation. First, we take the average duration per topic from ICSI, AMI, ELITR datasets (6 min/topic). For a topic to be considered as at least being touched upon, we want it to cover at least10% of the duration (~36s). Then we tested with a smaller internal dataset from our day-to-day meetings. The subjective evaluations from the meeting participants suggest that 30s is a reasonable threshold to determine if a topic is being touched at all.
>     * L120 (5min): This cut is only for the SIM recordings. For these recordings, the general flow is like this: First the moderator greets the participants and asks each participant to read out a sentence to test their audio device and network; then the discussion starts; at the end the moderator ends the meeting and may answer logistic-related questions from the participants.  Based on our subjective evaluations of the SIM recordings, the 5min cuts at the beginning and the end can safely remove the non-relevant contents (audio testing, logistic, etc).
> 4.	Regarding metrics: Yes, the F1 scores are available and we will add those to the outputs.
> 5.	Regarding anonymization: The SIM dataset marks all participants within a meeting as participant_1, 2, 3, 4 without referring to their real names. During the data collection, we collected user consent forms to use and distribute the collected data.
> 6.	Regarding availability: We anticipate all internal legal reviews to be done by September and can make the data and repo public by then.
> 7.	Regarding content improvements:
>     * Language and Typos: thank you so much for pointing out these. We will definitely update the language and typo issues (abstract, L267, L277, L288).
>     * Contents: Thank you so much for the suggestions. We will make the following updates to improve the contents: L123 – use “transcript” instead to reduce confusion; L265 – elaborate on why this is challenging; L274 – explain that we want to have people from the same industry (say legal) to discuss a topic in the domain and reproduce the type/style of meetings they would have; L266 – add short example or point to the Appendix.
>
> Thank you once again for your feedback and suggestions. If you have any other input, we are more than happy to address them accordingly.

---

### Official Review · Reviewer_n2Sz · 2024-07-27
**New task introduced for topic adherence in meetings; use of new and old datasets with multiple topics, gpt4 evals, weak metric**

**Rating:** 6
**Confidence:** 4

**Review:**

The paper presents a good resource for researchers for doing ‘topic relevance assessment’ for meeting transcripts. The dataset is diverse since it uses multiple existing datasets, including a new one the authors are releasing as well as synthetically generated data. In this research, it is not clear how the authors have defined ‘topics’ for the multiple datasets and how they have standardized these topics and annotations in various datasets they have selected. Authors highlight that they try to use topics not from post meeting summaries but pre-meeting agendas; however, authors do utilize the post meeting summaries for some of the annotations (as noted). The paper lacks detailed explanation of technical aspects, such as criteria for selecting meetings, the semantics of topical annotations for ground truth (are the annotations done automatically in the original datasets or manually with a focus beyond lexical overlap?).

Pros:
    - Comprehensive and diverse dataset covering various domains.
    - Use of SOTA LLM-as-a-Judge method (using GPT-4) for benchmarking
    - Open-source scripts for data augmentation, promoting reproducibility and further research.

Cons:
    - Use of Precision and Recall metrics may not be the most appropriate as topics may overlap, and having some semantic metrics or use of explainable methods for understanding the meaning of benchmark results is missing
    - Limited exploration of potential biases in GPT-4 evaluations.

**Strengths:**

One strength of this research may stem from the fact that the researchers are affiliated with Teams application that is used widely for online meetings. This perhaps gives the researchers some advantage for studying this topic with a broad perspective.

**Additional Feedback:**

None

**Clarity:**

Generally well written; however, see the issues noted in other sections that can be imrpoved to make the paper more readable.

**Correctness:**

Seems mostly correct; however since multiple datasets are aggregated and in many cases, and summaries or minutes are used as proxies for topics, it is not very clear to the reader how topics actually look like and whether they are consistent across all datasets and ground truth looks meaningful.

**Documentation:**

Looks fair.

**Ethics:**

No ethical concerns.

**Limitations:**

- SIM dataset does not look very natural and impromptu, only 4 participants create the dataset discussing 1 topic to generate the dataset. If the participants are researchers from the team, this might include various biases in the datasets.
- Non-verbal events are removed from the datasets (eg. ICSI) … this further removes the naturalness of the datasets.
- Use of synthetic meetings may not fully replicate the complexities of real-world conversations.
- The dataset’s focus on English-language meetings also limits its applicability in multilingual settings.

**Opportunities For Improvement:**

- Definition of Topic isn’t clear and properly defined. The benchmark metric (P vs R) looks naive and the scores are already pretty decent. The dataset could benefit from a more detailed exploration of potential biases and other issues besides topic. The paper could also delve deeper into the limitations and potential errors in machine-generated transcripts.

**Relation To Prior Work:**

The paper is related to the topic of meeting summarization, however this research focuses on pre-meeting agenda relevance (which authors say is somewhat novel task). It builds on prior datasets like the AMI and ICSI corpora but offers a unique contribution by combining these with newly collected data and synthetic meetings.

**Summary And Contributions:**

This paper introduces the Topic-Conversation Relevance (TCR) dataset, designed to assess the relevance of conversations to predefined meeting agendas. The dataset comprises 1,506 unique meetings, including both newly collected data and publicly available sources, such as the ICSI Corpus and MeetingBank. The TCR dataset provides a benchmark to evaluate the accuracy of GPT-4 baseline model in understanding the relevance of transcripts to meeting topics given the ground truth in the datasets. The contributions include compiling a multi-domain dataset, utilizing GPT-4 for rewriting detailed meeting minutes into pre-meeting agenda styles in an effort to standardize the topic annotations for evaluation.

---

> ### Author Rebuttal · Authors · 2024-08-16
>
> Dear Reviewer,
>
> We sincerely appreciate your time and feedback. In this response, we are providing additional details regarding your concerns and suggestions. Hope our responses can help address your comments.
>
> 1.	Regarding Metrics (Precision and Recall): we should have explained a bit better why we picked these metrics. We will add related contents to section 4.1. We picked these metrics from a feature point of view. Assuming that the LLM is a meeting moderator, we would like to have it remind the meeting participant that the current discussion is off-topic or behind schedule. To translate the feature into the benchmark problem, it is Precision (“LLM detects a topic is not being discussed and it is true”) and Recall (“A topic is not being discussed and LLM detects it”). In this case, good Precision makes sure that users get accurate notifications, and a high Recall guarantees that the feature can capture more cases where a topic is not being discussed.
> 2.	Regarding exploration of biases: This is a good suggestion, and we can add some lessons learned from (1) our prompt engineering process (2) current benchmark results. For (1), we can add some descriptions about prompt engineering techniques we used to help make the prompt more robust; for (2) we can add the raw prediction distributions across datasets to further demonstrate the differences across data sources.
> 3.	Regarding definition: We will definitely improve this by providing a clearer definition in section 1. Introduction. (Will also provide simple examples if the length permits.)
> 4.	Regarding SIM dataset: Sorry for the confusion, we will provide clearer explanations about the data. There is a better distribution split about the participants and demographic information from our previous work (Table 2 in Improving Meeting Inclusiveness using Speech Interruption Analysis). The TCR dataset includes a subset of the data used in the previous study. We will provide a breakdown of speakers as well in this paper’s Appendix. Thank you for raising this issue.
> 5.	Regarding Non-verbal events: We decided to remove the tags as we were thinking from the real-time transcript point view. For example, in most of the real-time transcribing services, laughter will be translated as “haha” instead of “tag laughter”; a coughing sound will not be transcribed instead of being marked as “tag no-verbal”.
> 6.	Regarding non-English language: Yes, this is actually something that we are working on. Based on the existing data, we used a non-GPT translating service to translate the transcripts and topics into other languages and run exactly the same tasks on them. So far we have tested Spanish and French. The results are very promising. We will definitely add them later when the TCR dataset once the more comprehensive results come in.
>
> Thank you once again for your feedback and suggestions. If you have any other input, we are more than happy to address them accordingly.

---

### Decision · Program_Chairs · 2024-09-26

**Decision:**

Accept (Poster)

**Comment:**

This paper forms a new meeting dataset, using existing meeting data and also collecting new ones. Reviewers made good suggestions, such as not removing the non-verbal events from the meeting, and the authors explained further in the rebuttals. We hope this information will be incorporated into the paper. This dataset will be helpful in reviving research on understanding meeting discussions.